# Lactadherin: From a Well-Known Breast Tumor Marker to a Possible Player in Extracellular Vesicle-Mediated Cancer Progression

**DOI:** 10.3390/ijms23073855

**Published:** 2022-03-31

**Authors:** Eduardo Durán-Jara, Tamara Vera-Tobar, Lorena De Lourdes Lobos-González

**Affiliations:** 1Centro de Medicina Regenerativa, Instituto de Ciencias e Innovación en Medicina, Facultad de Medicina Clínica Alemana, Universidad del Desarrollo, Santiago 7710162, Chile; eduranj@udd.cl (E.D.-J.); tvera@udd.cl (T.V.-T.); 2Advanced Center for Chronic Diseases (ACCDiS), Independencia, Santiago 8380000, Chile

**Keywords:** lactadherin, breast cancer, metastasis, extracellular vesicles, exosomes

## Abstract

Lactadherin is a secreted glycoprotein associated with the milk fat globule membrane, which is highly present in the blood and in the mammary tissue of lactating women. Several biological functions have been associated with this protein, mainly attributable to its immunomodulatory role promoting phagocyte-mediated clearance of apoptotic cells. It has been shown that lactadherin also plays important roles in cell adhesion, the promotion of angiogenesis, and tissue regeneration. On the other hand, this protein has been used as a marker of breast cancer and tumor progression. Recently, high levels of lactadherin has been associated with poor prognosis and decreased survival, not only in breast cancer, but also in melanoma, ovarian, colorectal, and other types of cancer. Although the mechanisms responsible for the tumor-promoting effects attributed to lactadherin have not been fully elucidated, a growing body of literature indicates that lactadherin could be a promising therapeutic target and/or biomarker for breast and other tumors. Moreover, recent studies have shown its presence in extracellular vesicles derived from cancer cell lines and cancer patients, which was associated with cancer aggressiveness and worse prognosis. Thus, this review will focus on the link between lactadherin and cancer development and progression, its possible use as a cancer biomarker and/or therapeutic target, concluding with a possible role of this protein in cellular communication mediated by extracellular vesicles

## 1. Introduction

### Lactadherin: A Milk Fat Globule Protein Associated with Cancer Development

Mammals produce milk fat globules (MFG) during lactation, which can contain up to 322 proteins, depending on the species [1]. About 50% of these proteins correspond to mammary tissue membrane proteins. Among them, lactadherin (also known as Milk Fat Globule-Epidermal Growth Factor 8 (MFG-E8), Breast Epithelial Antigen (BA46), SED1, among others) is a secreted glycoprotein associated with the milk fat globule membrane (MFGM) [1,2]. Human lactadherin consists of one N-terminal EGF-like domain which includes residues 24–67, followed by two repeated C domains (in human lactadherin, these include residues 70–255 and 230–387, respectively) [1]. The EGF-like domain contains an Arg-Gly-Asp (RGD) cell adhesion sequence motif that is recognized by integrin receptors, mainly αvβ3/β5 [3,4,5] (Figure 1). In humans, the *MFGE8* gene is located in chromosome 15, and codifies a 387 residues protein. After the cleavage of the signal peptide (residues 1–23), a 43–46 kDa full-length lactadherin protein, including residues between positions 24 and 387, is generated. There is also a short-length form of lactadherin which has a molecular weight of approximately 30 kDa, including residues between positions 202 and 387. This gives rise to an integral protein of the MFGM that has completely lost the EGF-like domain and some of the residues of the first C-domain [4,6].

The biological functions of each lactadherin form have not yet been completely elucidated. However, it is well known that lactadherin is present in the blood of lactating women, and that is highly expressed in breast tissue [1]. Several functions have been associated with its structural domains, mainly involving the interaction with αvβ3/β5-integrins present in macrophages and with phosphatidylserine (PS) exposed in apoptotic cells, therefore, promoting phagocytosis (efferocytosis) [7]. Other important functions described for lactadherin include antiviral properties [8,9], modulation of the innate immune system through the activation of tumor-associated macrophages [10], promotion of VEGF-induced angiogenesis [11], and tissue regeneration [12,13,14,15,16,17]. However, although lactadherin multifunctionality is still under discussion and analysis, most authors agree that lactadherin is projected as a strong focus of investigation, and a promising therapeutic target and/or biomarker for cancer and other diseases [10].

In 1991, Larocca and his group described that lactadherin is overexpressed in breast tumors, and that it has an RGD domain at the N-terminus [3]. This domain mediates cellular adhesion through the interaction with integrins. Therefore, it has been hypothesized to be part of the mechanisms by which lactadherin could exert its functions. This study also described lactadherin as a marker of breast cancer progression for the first time. Additionally, it demonstrated the presence of a soluble fraction, detectable in cancer patients, suggesting that lactadherin could be a suitable plasmatic marker of tumor progression [3]. A pro-tumorigenic function of lactadherin has been documented in several types of human cancers through largely undefined mechanisms. Thereby, it appears that lactadherin contributes to tumor progression [18,19,20,21], promotes survival of tumor cells, induces epithelial–mesenchymal transition (EMT) [2,18,21], and promotes angiogenesis [11,19], as well as metastasis [18,22]. However, the precise mechanisms are not completely elucidated (Figure 1). Moreover, recent studies have shown the presence of lactadherin in extracellular vesicles (EVs) derived from cancer cell lines, and in cancer patient specimens, which was associated with tumor aggressiveness and worse prognosis [23,24]. Thus, in this review, we will focus on the link between lactadherin and cancer development and progression, its use as a cancer biomarker and/or therapeutic target, concluding with a possible role of this protein in cellular communication mediated by EVs.

Schematic representation of lactadherin sequence and modeling prediction of lactadherin structure showing its three functional domains (center). The figure schematizes the N-terminus (blue) EGF-like domain (blue, containing the RGD motif), and the C1 and C2 domains at the C-terminus (light blue-green and green-red, respectively). Modeling prediction also shows the short length lactadherin, which includes mainly the C2 domain. Additionally, the postulated functions of lactadherin, both in normal physiology (left) and pathophysiology (right), are shown. Of note, it is still unknown whether full-length or short-length lactadherin have different biological roles. The modelling prediction was made using the Swiss-Model online tool (MFGM_HUMAN Q08431 Lactadherin).

## 2. Lactadherin as a Biomarker of Tumor Subtype, Progression and Metastasis

Although the expression of lactadherin was initially described in breast epithelial tissue, it has also been found in several other tissues and organs, especially in those related to the reproductive system, such as placenta, endometrium, and fallopian tubes [25,26,27,28,29]. One of the first reports of lactadherin as a tumor biomarker was published by Arklie et al. in 1981. They demonstrated that some antigens on the surface of mammary epithelium, whose expression change with the differentiation status, also were expressed in breast tumors [26]. However, at that time, lactadherin itself had not been identified, but rather a group of antigens, characteristic of the breast epithelium, was being discussed.

The most significant first advances regarding lactadherin as a tumor marker came from Ceriani and his group. They and others have extensively studied lactadherin expression in breast tumors, to develop monoclonal antibodies targeting lactadherin and other proteins of the MFG as immunotherapy against breast cancer [27,30,31,32]. This group detected lactadherin (more precisely, human epithelial antigens, among them is lactadherin) in the sera of patients with disseminated breast cancer [33]. After this intellectual–productive period of lactadherin-related research and its role in the biology and pathophysiology of breast cancer, there were many years without significant findings. Only since the mid-2000s, new information emerged regarding this protein involved in tumor development. Remarkably, during this time, its overexpression was also described in other types of cancer, including pancreatic cancer models [19], bladder cancer [20], ovarian cancer [34], oral cancer [35], among others [36,37,38] (Table 1).

Several other articles identifying lactadherin as a tumor marker were seen in melanoma models and patients. In 2008, Jinushi et al. showed, for the first time, lactadherin overexpression in the tumor growth phase in a murine model of aggressive melanoma [39]. A few years later, its overexpression was shown in both primary tumors and metastases. Furthermore, in this study, its overexpression was associated with a poor prognosis and survival [40]. On the other hand, in 2016, an interesting study showed that, in addition to acting as a tumor tissue marker, lactadherin was present differentially around blood vessels, especially in pericytes, probably participating in the angiogenesis of the tumor [41]. This was shown in an in vivo murine model and in melanoma patients. In addition, an important role of lactadherin expressed in mesenchymal stem cells (MSCs) in the development of this type of cancer was shown, mainly increasing VEGF and ET-1 expression, and enhancing M2 polarization of macrophages, thus promoting tumor angiogenesis [41].

Additionally, in 2017, another study suggested lactadherin as a possible biomarker for colorectal cancer (CRC). Using open-access TCGA data, Zhao et al. observed lactadherin overexpression in advanced CRC, compared to early stages of the disease and adjacent healthy colonic tissue. Similarly, its overexpression was correlated with different metastasis parameters and a poor prognosis [42]. Almost simultaneously, another research group showed lactadherin overexpression (this time, also at the protein level) in tumor tissue versus adjacent healthy tissue in two different cohorts of Chinese patients [18]. In accordance, its overexpression was associated with metastasis, and correlated with parameters of poor prognosis (Table 1). Recently two different studies showed the importance of lactadherin in hepatocellular carcinoma (HCC) diagnosis and progression. The first study by Shimagaki et al. showed that serum lactadherin levels may be used as a prognostic or diagnostic biomarker in HCC patients [43]. However, in contrast with other reports, patients with this type of cancer displayed lower levels of serum lactadherin compared with patients with other inflammatory liver diseases and healthy controls. Interestingly, serum lactadherin levels can distinguish HCCs, even those negative for α-fetoprotein (AFP) or des-γ-carboxy prothrombin (DCP) in patients with liver cirrhosis or chronic hepatitis [43]. On the contrary, but according with most of the literature, the second study showed that lactadherin is overexpressed in HCC patient biopsies compared with healthy liver tissue. They also showed that lactadherin overexpression promotes migration and invasion of HCC cell lines in vitro, and enhanced tumor growth in a xenograft murine model of HCC [44], demonstrating that lactadherin plays an important role in HCC progression.

On the other hand, after numerous studies regarding lactadherin’s role in breast cancer development and progression in the 80s and 90s, they suffered a major break. The use of lactadherin as a possible tumorigenic marker in this type of cancer reappeared in 2012. Carrascosa et al. analyzed gene expression public data (microarrays), and showed lactadherin overexpression both in primary tumors and in breast cancer metastases. Interestingly, its expression was associated with the absence of hormonal receptors [21]. Other studies have also associated lactadherin expression levels with the absence of hormonal receptors, being higher in triple negative cell lines and in triple-negative breast cancer (TNBC) biopsies [45,46]. After another long period, recently in 2019, the expression of this protein was analyzed in a cohort of Chinese patients, and its association with different clinic-pathological and prognostic parameters was investigated. The study evidenced higher lactadherin expression in tumor tissue versus adjacent healthy tissue, and found an association of lactadherin expression with relevant clinical and prognostic factors, such as TNM stage, lymph node metastasis, and reduced survival [46]. Moreover, another recent study showed the presence of a recurrent gene fusion comprising MFGE8 and HPLN3 genes using TNBC patient genomic data. This gene fusion was much more frequent in TNBC than in other subtypes. The authors further validated the presence of the MFGE8–HPLN3 gene fusion in TNBC biopsies, and predicted the length of the encoded protein, suggesting its critical role in TNBC biology, and a possible clinical application as a new biomarker [47]. In Table 1, we listed the main discoveries of lactadherin research regarding cancer development and its importance as a biomarker of this disease.

## 3. Lactadherin Role as Promotor of Tumor Progression

Lactadherin expression has been associated with the aggressiveness and progression of several types of cancer. In fact, most of the studies associated lactadherin overexpression in tumor samples with the promotion of pro-tumorigenic and pro-metastatic capacities, such as increased tumor cell proliferation, angiogenesis, migration, invasion, and epithelial-to-mesenchymal transition (EMT) [11,19,21,22,39,40,41,42,48]; processes that are related to the physiological function of this protein. In this context, several studies in the cancer field, but also in other diseases, have demonstrated the activation of commonly pro-tumorigenic signaling pathways mediated by lactadherin overexpression, such as their interaction with β3-integrin, and the activation of the PI3K/AKT signaling [39,42,49,50,51]. On the other hand, most of the studies have not deepened in the downstream intracellular signaling triggered by lactadherin, thus leaving the field open for more complete and comprehensive studies.

The investigation of lactadherin function in tumor development and progression is an important topic in order to understand the mechanisms underlying these processes, especially in the search for better and new therapeutic approaches. However, as far as we know, those mechanisms have not yet been fully elucidated. The available studies show that lactadherin has multiple effects on tumor development, consistent with its various biological and physiological functions. These effects can be separated mainly in two different ways: (i) functioning as an adhesion molecule, or (ii) participating as a signaling molecule, triggering intracellular signaling pathways. In both cases, lactadherin can promote cell survival and proliferation, EMT, immunomodulation, and angiogenesis [1] (Figure 1). Here, we briefly summarize the role of lactadherin in tumor cell proliferation/survival, EMT, as well as its immunomodulatory function.

### 3.1. Tumor Cell Survival/Proliferation and EMT

As described earlier, lactadherin is able to interact with αvβ3/β5-integrins through its RGD domain. This interaction allows the anchorage and rapprochement between cells (or extracellular matrix), thus favoring apoptotic cell clearance through phagocytosis [2,5,7]. On the other hand, the activation of β3-integrin signaling can lead to the activation of pro-survival, anti-apoptotic, and pro-metastatic pathways through the activation of molecules such as Akt and Twist, as shown in murine melanoma studies [39]. This lactadherin/β3-integrin interaction could promote the EMT process (through PI3K/Akt, Twist); an effect that has also been observed in CRC cancer models, where lactadherin promotes the migration and invasion of tumor cells [42].

In an interesting study using *Mfge8* KO mice, the authors used two different models of colon cancer, and evaluated the role of lactadherin and αv-integrin in epithelial and tumor cell proliferation. They showed that in both colitis-induced and sporadic models of colon cancer, tumor size was lower in KO mice as compared to WT mice. In vitro findings showed that lactadherin promotes epithelial cell proliferation. Moreover, treatment with an siRNA targeting αv-integrin reduced the proliferation of CT26 colon cancer cells stimulated with recombinant lactadherin, indicating that growth and proliferation of epithelial and tumor cells are controlled, at least in part, by the interaction between lactadherin and αv-integrin [52]. Another recent study in CRC was done by Cao et al. They showed that lactadherin is overexpressed in CRC biopsies compared with colon adjacent healthy tissue. They also showed that the downregulation of lactadherin by administering coptisine results in in vivo tumor growth suppression, decreased adhesion, and metastasis of CRC cells. Coptisine administration also inhibited the expression of MMP-2 and MMP-9 via the PI3K/AKT signaling pathway, and inhibited EMT both in vivo and in vitro [53].

On the other hand, it has been shown that in highly aggressive basal/TNBC subtype cells, lactadherin expression is regulated by p63 protein, which functions as a pro-survival and pro-tumorigenic factor. On the contrary, in estrogen receptor (ER) and erbB2-positive breast cancer cells, which may have better prognosis, lactadherin exhibits tumor suppressive functions, and is not regulated by p63 [45]. The work by Carrascosa et al. explored the role of lactadherin in breast cancer development and its association with the increase of the tumorigenic potential of tumor cells. They found that overexpression of this protein increases aggressiveness and tumorigenicity of breast tumor cells. In contrast, lactadherin knockdown decreased tumor growth. Such modulation of tumor growth and its aggressiveness was mediated by the activation of cyclins D1/D3 and N-cadherin, suggesting the participation of lactadherin as a promoter of proliferation and EMT [21]. Recently, lactadherin expression has been shown to be associated with the induction of apoptosis and the expression of EMT markers in breast cancer cell lines, leading to modulation of the migration and invasion capabilities of tumor cells [48].

### 3.2. Lactadherin Immunomodulatory Role

Another important function of lactadherin is its immunomodulatory role, mainly as an immunosuppressive molecule [1]. The first reports in this topic show that lactadherin is able to bind β3-integrin on the surface of phagocytes, mainly macrophages. In the same way, lactadherin can bind PS exposed in cells undergoing apoptosis, and promote their phagocytosis [7] (Figure 1). Subsequent studies have shown that other cells of the immune system also express lactadherin, such as dendritic cells, mast cells, and regulatory T lymphocytes, in addition to endothelial cells that are in direct contact with blood and blood cells [1,11].

One of the first reports on the participation of lactadherin and its immunomodulatory function promoting the tumorigenic process dates from the mid-2000s. In this study, lactadherin was shown to promote the formation of bladder tumors in immunocompetent mice [20]. On the other hand, using a carcinogen-induced bladder cancer model in immunosuppressed *Rag2* KO mice, these animals did not develop advanced tumors despite expressing lactadherin, thus evidencing a strong link between lactadherin and the adaptive immune system. Additionally, a correlation was observed between lactadherin levels and the number of tumor-associated macrophages and infiltrating regulatory T lymphocytes in human samples, suggesting a poor prognosis for these patients [20]. Another study from 2011 showed the importance of lactadherin expressed by tumor infiltrating macrophages in cancer stem cell function regulation. This effect had two main consequences: promoting tumorigenicity and drug resistance [10].

The immunomodulatory role of lactadherin was also observed in murine models of melanoma. Using siRNA to silence lactadherin expression, a significant decrease in tumor growth was observed, which was accompanied by decreased tumor neovascularization and decreased levels of infiltrating regulatory T lymphocytes [54]. Similarly, Yamada’s group demonstrated the involvement of lactadherin present in MSCs in the development of melanoma using B16 tumor cells. Furthermore, lactadherin was observed to be expressed at higher levels in MSCs versus B16 tumor cells. When B16 and MSCs cells were co-injected into mice, it was observed that wild type MSCs promoted the growth of B16 tumors when compared with the co-injection of MSCs that do not express lactadherin [41]. Moreover, the authors showed that the increase in tumor size was associated with an increase in angiogenesis and macrophage polarization to an M2 phenotype. In the same study, a higher presence of lactadherin around blood vessels within the tumor was also observed in human melanoma samples, which could suggest its participation in tumor angiogenesis [41] (Figure 1 and Table 1).

Lactadherin has also been involved in the modulation of the immune system in other types of cancer, such as oral cancer [35], esophageal cancer [36], and, recently, angiosarcoma [37] and a murine glioma model [55]. The immunomodulation in these types of cancer have been constantly associated with tumor development and bad prognosis. Also, these studies mainly show a correlation between high levels of lactadherin in tumor biopsies with high levels of M2 macrophages, regulatory T lymphocytes, and low levels of infiltrating CD8 T lymphocytes.

## 4. Lactadherin as a Possible New Cancer Therapeutic Target

Although lactadherin has various regulatory physiological roles, and its expression and function have been targeted for the treatment of several other diseases (e.g., vascular and autoimmune pathologies) [13,14,15,51,56,57,58], it has been established to be most relevant in the oncologic area. First studies by Ceriani et al. showed that an antibody cocktail, including MoAbs against HMFG components (one of them against lactadherin), worked as treatment and prevention against the engraftment of ER-positive and negative mammary tumors [27,30]. Later, this group and others published several reports using this cocktail of antibodies, including an anti-lactadherin blocking antibody, demonstrating its beneficial effect against breast tumors [31,32,59]. Subsequently, several studies including some reviewed here, reported lactadherin as a possible cancer therapeutic target [20,21,22,41,48], either using gene therapy approaches, inhibiting its expression by siRNAs or shRNAs, or by using specific antibodies that block its function and interactions with αvβ3/β5 integrins [45] (Table 1).

Recently, using TCGA data and machine learning, Kothari et al. have shown that lactadherin is one out of two proteins/genes that can differentiate TNBC from non-TNBC irrespective of their heterogeneity or subtype differences [60]. Further affinity purification mass spectrometry and proximity biotinylation experiments identified a possible role for lactadherin in various tumor survival processes. Another study also using TCGA RNA-seq data showed that lactadherin, as well as KLK5/7 expression, are associated with COX-2 inhibitors treatment resistance in TNBC cells. This study demonstrated that silencing of these genes markedly recovered COX-2 inhibitor sensitivity both in vitro and in vivo. Considering the difficulty in the treatment of TNBC, these results could support the possibility of using new combination therapies against TNBC involving COX-2 and lactadherin inhibition [61].

### State of Art on Lactadherin Translational Medicine Research and Intellectual Property

When it comes to intellectual property, there are several patents that postulate the use of lactadherin. A few of them postulate this protein as a biomarker for different types of cancer, or as a marker of metastasis within an array of other proteins and molecules. On the other hand, to date, there are several important patents that include the use of lactadherin as a therapeutic target in their approach. One of them proposes the treatment of different cancers using anti-lactadherin antibodies (US9226934B2). A second one proposes it as a radio-protective molecule (US20150080304A1). The third one proposes the use of a lactadherin cyclic peptides (cLacs) as a possible early indicator of apoptosis (US20140194369A1). Besides these, a Korean patent proposes combinations of peptides as immunotherapy for treating ovarian and other cancers (KR20180022968A). This approach is similar to what US researchers proposed back in 1999: tumor-associated antigen peptides (e.g., lactadherin peptides) can be used as anti-tumor vaccines (WO2000006723A1). There is another patent from 2013 where the use of lactadherin-blocking antibodies is proposed for the treatment of different diseases, including some types of cancer (WO2013139956A1). Finally, a recent patent postulates the detection of lactadherin, among another two proteins (CA153 and CEA), as tumor markers for several types of cancer. It considers a method to detect these proteins in blood samples of cancer patients (CN106596940A). It proposes that the combined detection of these three proteins can enhance test sensitivity, specificity, and precision.

Strikingly, despite the different patents and therapeutic approaches, to date, lactadherin has not been a focus of investigation in clinical trials. Interestingly, based on studies by Touhy and collaborators (Mazumder’s group) [62,63], a phase I clinical trial began the enrollment of subjects to test safety, as well as the most effective dose of an alpha-lactalbumin vaccine to treat patients with non-metastatic TNBC, in October 2021. As lactadherin, alfa-lactalbumin is not expressed in normal tissues at immunogenic levels (only in lactation), but is overexpressed in emerging tumors. Currently, this clinical trial is recruiting patients, and the estimated primary completion date is May, 2022 (https://clinicaltrials.gov/ct2/show/NCT04674306 accessed on 2 March 2022).

Finally, there are a few patents that include the use of lactadherin associated with patient’s EVs, which implies the use of liquid biopsies, which are samples of much easier access. For example, the Japanese patent JP2016028572A considers the use of lactadherin (among an array of other proteins) present in EVs from different bodily fluids to characterize a patient’s phenotype or condition. On the other hand, another recent patent (CA2988771A1) includes the use of lactadherin to purify a subpopulation of EVs, based on the interaction and binding between this protein and phosphatidylserine (PS) present on EVs’ surface. As mentioned before, the field of circulating EVs is interesting because the use of blood or body fluids samples is much easier than the acquisition of solid tumor biopsies, and it can take the advantage that EVs cargo can reflect that of the tumor origin cell, so they can serve as biomarkers of the disease.

## 5. Extracellular Vesicles and Exosomes as Promotors of Breast Cancer Metastasis

Metastasis is defined as the dissemination of cancerous cells from the primary tumor, and the effective colonization of secondary target organs. It is widely accepted that intercellular communication is essential in all steps of the metastatic cascade. Exosomes are a particular subpopulation of EVs released by a variety of cell types. These exosomes are 40–200 nm in diameter, and are derived from the multivesicular endosome pathway, and can enter a recipient cell mainly through three different pathways: membrane fusion, endocytosis, or the interaction of proteins in exosomes with receptors in recipient cells [64,65,66,67]. Exosomes are thought to play important roles in intercellular communication, transferring a variety of molecules to target recipient cells. Exosomes contain several bioactive molecules, such as nucleic acids (mRNA, microRNA, DNA, and other non-coding RNAs), proteins (receptors, transcription factors, enzymes, extracellular matrix proteins), and lipids that can redirect the phenotype and function of a recipient cell [67,68,69,70]. Therefore, exosomes are emerging as local and systemic cell–cell mediators of oncogenic information that play an important role in cancer progression [65,66,67,68,69,70,71,72].

In breast cancer, there is a potential use of exosomes and other EVs as promising diagnostic and therapeutic biomarkers. On the other hand, exosomes secreted by metastatic cells potentiate tumorigenic capacities of less aggressive cells acting in a paracrine manner [68,69,70,71,72,73]. Moreover, recent reports indicate that exosomes are able to condition the microenvironment, where tissue recipient cells are waiting for the advent of a tumor cell, and which, in turn, are capable of promoting angiogenic mechanisms in breast cancer [71]. At the same time, exosomes secreted by tumor stroma can also influence tumor progression. Breast-cancer-associated fibroblasts secrete exosomes that have been shown to promote tumor mobility, invasion, and dissemination of breast cancer cells through the Wnt-planar cell polarity pathway (Wnt-PCP pathway) [74].

Many research groups worldwide have focused on the characterization of exosomes and their functionality using in vitro assays. In breast cancer, several researchers have studied the contribution of exosomal microRNAs and their role in metastasis development [75,76,77,78]. However, very few researchers have been able to describe specific proteins contained in exosomes that are able to prepare metastatic niches in tissues. In this regard, there is one publication that addresses, in part, the aforementioned points. This study from 2015 led by Leyden’s group showed that tumor-derived exosomes up-internalized by organ-specific cells prepare the pre-metastatic niche [72]. Remarkably, exosome proteomics revealed distinct integrin expression patterns, in which the exosomal integrins α6β4 and α6β1 were related to lung metastasis, whereas exosomal integrin αvβ5 was linked to liver metastasis. These findings strengthened research in the area of exosomes as promoters of metastasis.

## 6. Role and Use of Lactadherin in EVs and Exosomes

Lactadherin expression has been reported in a variety of tissues and organs. However, its overexpression has been associated with bad prognosis and outcomes in different types of cancers (Table 1). Moreover, lactadherin presence in EVs has recently been reported in several studies and databases [79,80,81], which has led to the consideration of it as a possible new marker of EVs. On the other hand, lactadherin C1C2 domains have been extensively used in EV designing and engineering. Due to its presence in EVs, and its interaction with PS (also present on EV surfaces), these domains are usually fused to a protein of interest, thus directing target protein to the EV surface [82,83,84,85]. This strategy has also been used to engineer EVs to expose anti-HER2 scFv on their EVs surface, and redirect EVs to HER2+ breast cancer cells to deliver a cargo of mRNA gene therapy [86]. More recently, a similar strategy was used by Kooijmans et al. to decorate EVs with EGFR-specific nanobodies fused to the C1C2 domains of lactadherin to further improve tumor cell targeting and incorporation [87]. The authors hypothesize that this effect is mainly mediated by the interaction of lactadherin C1C2 domains with PS present on the EV surface, thus protecting EVs from recognition by plasma components and phagocytes, preventing or retarding their clearance form circulation. On the other hand, using bioluminescent and fluorescent labeling of human and mouse-derived EVs, an interesting recent work by Gupta and collaborators showed that the proportion of lactadherin+ EVs is significantly lower than that of tetraspanin+ EVs. Transfecting human embronic kidney HEK-293T cells with fusion construct (lactadherin-GFP or tetraspanin-GFP) and subsequent sorting of the EVs based on GFP showed a high percentage of tetraspanin+ EVs (15–25%). In contrast, just 1% of the sorted EVs was lactadherin+, which could have implications on EVs isolation or affinity purification protocols [88]. However, it is important to mention that the authors check this initial aproach only in HEK293T cells, which is a non-tumor cell line. It is possible that, as mentioned before, lactadherin is more abundant in tumor or immune cell-secreted EVs, according with its related functions.

With all of this in the background, now the questions are, on one hand, is lactadherin present in exosomes or EVs secreted by tumor cells? And, if it is, does lactadherin have a pro-tumorigenic role in these EVs? The literature suggests that lactadherin can have a scaffolding role to anchor EVs to their recipient cells or to extracellular matrix components [89]. One study describing the presence of lactadherin associated with EVs in a kidney cell line suggested a possible role of this protein in membrane secretion processes, such as budding or shedding of plasma membrane (microvesicles) and endocytic multivesicular bodies (exosomes) [90]. Other reports have shown lactadherin presence in EVs secreted by dendritic cells, suggesting an immunomodulatory role [81]. Particularly in the case of breast tissue (normal or tumor), there are few studies describing the presence of lactadherin associated with exosomes or EVs. One of these articles indicates that lactadherin in EVs is required to transduce cellular signals from the basolateral side of adherent cells by accumulating exosomes in mammary epithelial cells [89]. However, this report did not describe the presence of lactadherin on the exosomes. Finally, recent studies by our group described the presence of lactadherin as part of the characterization of exosomes secreted by a metastatic breast cancer cell line (MDA-MB-231); however, functional experiments studying its specific and precise role were not performed [23,91]. As lactadherin has several functions, both as a cell adhesion molecule, and triggering intracellular signaling cascades, it is possible to think that it could have similar roles as part of EVs membranes of EVs cargo. Figure 2 schematizes both the known role of lactadherin as an immunomodulatory molecule as part of the cell membrane of phagocytes and its possible role as a component of EVs’ membranes, where it could function mediating EV-cell interaction and internalization, and triggering intracellular signaling pathways, mainly through the interaction with αvβ3/β5 integrins (Figure 2).

**Table 1 ijms-23-03855-t001:** Main highlights in cancer-related lactadherin research.

Cancer Type	Experimental Strategy	Findings	References
Breastcancer (BC)	Hybridoma clones to obtain MoAbs against HMFG components	MoAbs binds to human epithelial BC cell lines, breast epithelium sections of primary carcinomas and lactating breast. Not reactive against epithelial cell lines of non-breast origin	[26,30]
	Antisera against HME-Ags to determine its presence in the sera of patients with disseminated BC and others	HME-Ags detected in sera of BC patients, but not in sera from non-BC patients or in normal female controls	[33]
	Human mammary epithelial cells to characterize the specificity of MoAbs against HMFG components	Determination of the molecular weight of three different targets of MoAbs. One of them is directed against a 46 kDa protein	[27,30]
	MoAbs against Ags of the HMFG to characterize different BC cells	MoAbs are useful in characterizing breast epithelial cells, studying surface alterations in malignancy, and possibly in BC diagnosis and therapy	[32]
	MoAbs to select complementary DNAs from an expression library of lactating breast to better characterize the 46 kDa antigen	Detection of one large complementary DNA, encoding a 217 aa peptide. Detection and OE of a single 2.2 kb RNA in a variety of carcinoma cell lines. Sequencing revealed strong homology of the 46 kDa glycoprotein with serum factors VIII and V	[3]
	Purified lactadherin to evaluate RGD-dependent cell adhesion	Lactadherin promotes RGD-dependent cell adhesion. This is dependent of integrins, mainly αvβ3	[2]
	Immortalized mammary epithelial cells Analysis of public gene expression data	Lactadherin promotes tumorigenic potential through regulation of cyclins D1/D3 and N-Cadherin Lactadherin is highly expressed in primary and metastatic BC, associated with absent ER expression	[21]
	IHC to evaluate lactadherin presence in BC biopsies	Lactadherin OE in BC tissue. OE associated with poor prognosis parameters	[46]
	Analysis of TCGA patient genomic data	Detection and validation of MFGE8-HPLN3 in TNBC patient specimens	[47]
	EVs secreted by MDA-MB-231 human BC cells. WB analysis	EVs secreted by MDA-MB-231 cell line contain lactadherin	[91]
	Proteomic and WB analysis	Exo-WT (secreted by wild type MDA-MB-231) are enriched with lactadherin compared with Exo-1537S Exo-WT (containing lactadherin) promotes tumorigenic capacities on recipient cells and mice	[23]
Pancreatic cancer	Transgenic Rip2-Tag2 mouse model of multistage carcinogenesis	Lactadherin OE in angiogenic islets and tumors of mice compared with normal pancreas, promoting tumor growth	[19]
Melanoma	Murine B16 melanoma cells Human primary melanoma cell lines	High expression of lactadherin in the growth phase of melanoma, promoting melanoma progression through Akt/Twist signaling	[39]
	IHC of primary melanoma, metastatic lesions and benign tissue.	High expression of lactadherin in primary and metastatic melanoma High expression associated with tumor progression and worse survival	[40]
	Co-injection of B16 melanoma cells and MSC derived from wild type (WT) or *MFGE8* KO into mice	Lactadherin promotes melanoma growth through MSCs-induced angiogenesis and M2 polarization of TAMs	[41]
Bladder cancer	Transcriptomic analysis of bladder carcinoma biopsies	Lactadherin OE during tumor development Correlation between lactadherin levels with expression of genes involved in cell adhesion, migration, and immune response, promoting tumor growth	[20]
Ovarian cancer (OC)	IHC of human OC biopsies Lactadherin blocking MoAbs effect over tumorigenic capacities of human OC and TNBC cell lines	Lactadherin OE in OC biopsies and in a TNBC cell line Blocking MoAbs efficiently blocked lactadherin tumor-promoting effects, such as survival, migration, and adhesion	[22]
	Lactadherin and CD133 expression levels were analyzed by IHC in epithelial OC specimens	Lactadherin OE significantly correlated with the presence of CD133 Lactadherin and CD133 levels significantly correlated with bad prognosis parameters in OC	[34]
Colorectal cancer (CRC)	IHC and IF analysis of lactadherin expression in CRC biopsies	Lactadherin OE in CRC samples compared with normal mucosa tissues Lactadherin in close proximity to endothelial cells. Expression correlated with bad prognosis parameters and worse survival	[18]
	IHC and qRT-PCR analysis of lactadherin expression in CRC biopsies Analysis of CRC RNAseq data from TCGA Use of shRNA and recombinant human lactadherin to investigate its role in CRC cell growth, migration, and invasion	Lactadherin OE in advanced CRC tissues compared with early-stage CRC and adjacent non-cancerous tissues Lactadherin levels correlated with bad prognosis and survival parameters Lactadherin promotes CRC cell migration and invasion, increases levels of MMP-2 and MMP-9, and promotes EMT and promotes progression via AKT/MMPs signaling	[42]
	Evaluation of MFG-E8 levels in patient’s tumor specimens Evaluation of antitumor effect of coptisine in vitro and in vivo	OE of MFG-E8 in human CRC tissue samples versus adjacent normal ones Coptisine inhibited CRC growth and progression by downregulating MFG-E8 expression, and inhibiting EMT and expression of MMP-2 and MMP-9 via the PI3K/AKT signaling pathway	[53]
Gastric cancer (GC)	Analysis of GC RNAseq data from TCGA	MGFE8 mRNA level associated with worse survival Establishment of a prognostic predictive model for GC that includes *MFGE8* gene measurement	[38]

Abbreviations: MoAbs: Monoclonal Antibodies; Ags: Antigens; OE: Overexpression; HMFG: Human Milk Fat Globule; RIA: Radioimmunoassay; HME-Ags: Human Mammary Epithelial Antigens; ER: Estrogen Receptor; EVs: Extracellular Vesicles; WB: Western Blot; IHC: Immunohistochemistry; MSCs: Mesenchymal Stem Cells; TAMs: Tumor Associated Macrophages: IF: Immunofluorescence; TCGA: The Cancer Genome Atlas; MMPs: Mielometaloproteinases.

On the other hand, there are just a few papers that describe the presence of lactadherin in exosomes from cancer patients. Soki et al. showed higher levels of this protein in circulating exosomes from metastatic prostate cancer patients compared with non-metastatic cancer patients or healthy controls [92]. Also, in prostate cancer, using a proteomic approach, Clark et al. showed that lactadherin is differentially present in some subpopulation of EVs, depending on the presence of FUT8 protein [93]. Another recent study showed the presence of lactadherin in circulating exosomes from hepatocellular carcinoma patients. Interestingly, the correlation showed in this type of cancer (in this study) was inverse [43]. The authors hypothesize that an interaction between PS on EVs’ surface and serum lactadherin exist. They showed that lactadherin levels are almost undetectable in patient’s EV-depleted sera, and that HCC patients have decreased levels of serum EVs, thus explaining their results. Notably, a recent study showed that lactadherin is selectively present in distinct subpopulations of melanoma-derived EVs, being enriched in small low-density EVs, among other proteins, such as EHD1 and EHD4, PTGFRN, RAB1A, ADAM10, and ALIX [24]. Similar results were obtained by Temoche-Diaz et al. using MDA-MB-231 metastatic breast cancer cells [94]. Figure 3 summarizes the principal discoveries (milestones-highlights) about lactadherin research, focused on its role in tumor progression and metastasis, including research related with its presence and role in EVs (Figure 3).

## 7. Conclusions

Lactadherin is present widely in human tissues, and can easily be targeted by its special multi-domain structures. Lactadherin can promote tumor formation, and prompts cancer vascular angiogenesis, survival, and EMT, regulating multiple oncogenic pathways (p63/p73, PI3K/Akt, β-catenin, Akt/Twist), which can promote cancer cell resistance to chemotherapy and host immunity suppression. However, to date, the specific roles, molecular mechanisms, and signaling pathways by which lactadherin promotes tumorigenic and metastatic properties of tumor cells, especially breast cancer cells, are not completely elucidated. Despite the interesting fact that several patents have used lactadherin as a possible tumor marker or even therapeutic target, there are no reported clinical trials in which this protein is being tested. On the other hand, it is well known that EVs and exosomes can mediate cell–cell communication, and can promote the acquisition of oncogenic and pro-metastatic properties in recipient cells. Past and recent works have shown the presence of lactadherin in EVs secreted by different cell types. However, the role of lactadherin present in EVs, more specifically, in exosomes secreted by cancer cells (or other cells in the tumor microenvironment), is still unknown. Given the importance of this protein modulating tumor development and progression, deeper understanding on the role of lactadherin in EV- and exosome-mediated tumor progression and metastasis could be a promising focus of study in the future.

## Figures and Tables

**Figure 1 ijms-23-03855-f001:**
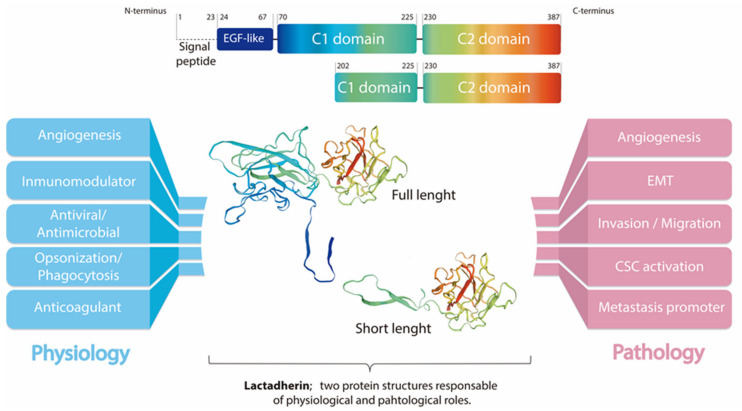
Lactadherin structure and possible roles in physiologic and pathologic states.

**Figure 2 ijms-23-03855-f002:**
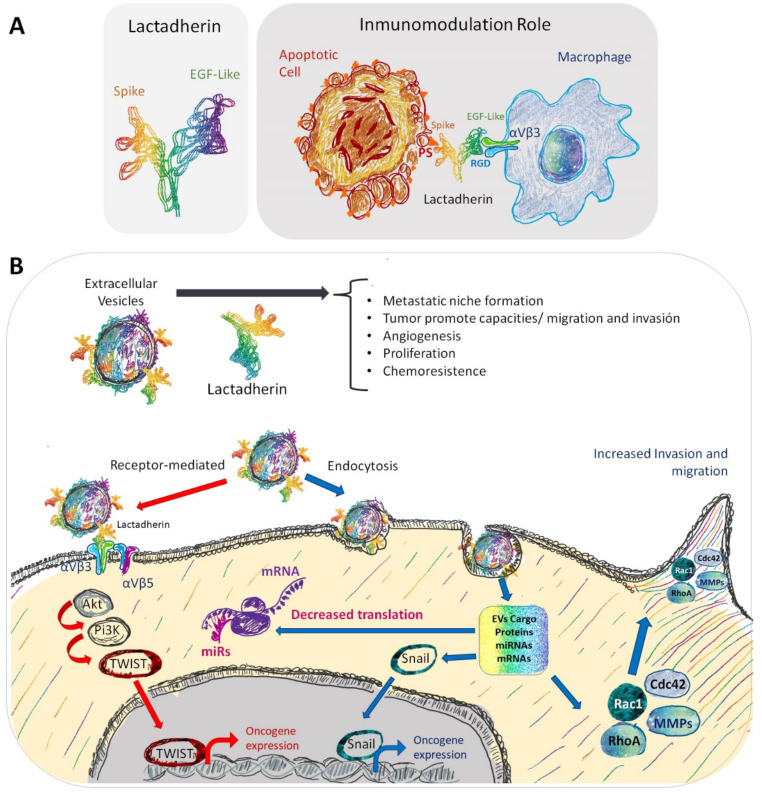
Representation of the immunomodulatory role and other possible roles of lactadherin present in EVs. (**A**) In non-lactating conditions, epithelial cells have low basal lactadherin expression. (**B**) However, in a tumor context, lactadherin overexpression is correlated with tumor progression through several ways, one of them being its presence in EVs. In this EV setting, lactadherin may be promoting tumor cell survival, migration, and invasion capacities, promoting angiogenesis and EMT. Also, lactadherin presence on EVs membranes could have a role as a targeting ligand for EVs towards recipient cells.

**Figure 3 ijms-23-03855-f003:**
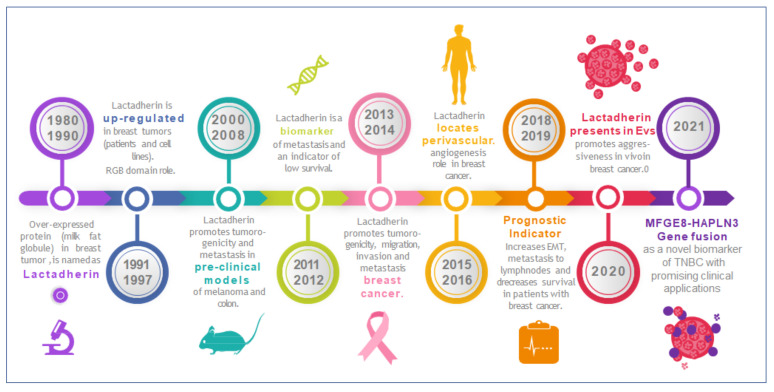
Timeline highlighting lactadherin research on its role in tumor development, progression, and metastasis. Lactadherin was first associated with breast tumors in the mid-80s to the early-90s. Since then, periods of fruitful research have been identified roles in tumor growth, progression, and metastasis. Lately, recent investigations have also identified lactadherin in EVs and exosomes, proposing it as a promising biomarker and/or therapeutic target.

## Data Availability

The datasets analyzed and reviewed during the current study are available in the public open access databases, UCSC Xena and Uniprot repositories, references [25,26].

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
