# Peer review of "Lactadherin: From a Well-Known Breast Tumor Marker to a Possible Player in Extracellular Vesicle-Mediated Cancer Progression"

_ijms, 2022, doi:10.3390/ijms23073855_

Round 1

Reviewer 1 Report

Comments to authors

The manuscript by Durán-Jara et al., “Lactadherin: from a well-known breast tumor marker to a possible player in extracellular vesicle-mediated cancer progression” is dedicated review to the lactadherin – promising target for cancer therapy.

Scrupulous checking of the references used in this manuscript, there are few comments/suggestions mentioned below.

One main comment about style – to look again at the manuscript and minimise use of word “interesting study” or “interestingly”. There are just a few places where it is not necessary.

In addition, please, revise the references on figures in the text. Some unneeded (e.g. Fig 1. mention is not needed in Introduction paragraph 2 in sentence “However, although lactadherin multifunctionality is still under discussion and analysis (Fig. 1), most authors agree that lactadherin is projected as a strong focus of investigation and a promising therapeutic target and/or biomarker for cancer and other diseases [10].”) or not in the order (e.g. Fig.3. is mentioned in the text ahead of Fig.2. – revise, if it is really needed).

1.Introduction

Comment 1: “Other important functions described for lactadherin include antiviral properties, modulation of the innate immune system through the activation of tumor associated macrophages [8], promotion of VEGF-induced angiogenesis, and tissue regeneration [9,10,11-15].”

Could not find in given references statement about antiviral properties.

About VEGF-induced angiogenesis authors gives reference 12 and 13, however, both are just mentioning original paper by [20] - Silvestre JS, Thery C, Hamard G, et al. Lactadherin promotes VEGF-dependent neovascularization. Nat Med 2005;11:499, hence, please, review necessity for reference 12. and 13.

  1. Matsuda, A. et al., Milk fat globule EGF factor VIII ameliorates liver injury after hepatic ischemia-reperfusion. Journal of Surgical Research 2013,

180, Pages e37-e46.

  1. Deng, Ke-Diong. et al., Restoration of Circulating MFGE8 (Milk Fat Globule-EGF Factor 8) Attenuates Cardiac Hypertrophy Through Inhibition

of Akt Pathway. Hypertension 2017, 70, 770–779.

In addition, reference 14 is about mesenchymal stem cells. Yes, in this review, there is a reference to Motegi et al., original paper about VEGF relation to MEG8. Please, cite the original paper.

  1. Motegi, S.; Ishikawa, O. Mesenchymal stem cells: The roles and functions in cutaneous wound healing and tumor growth. Journal of Dermatological Science 2017, 86, 83-89.

Comment 2: Statement “Moreover, recent studies have shown lactadherin presence in extracellular vesicles (EVs) derived from cancer cell lines and in cancer patient specimens, which was associated with tumor aggressiveness and worst prognosis” – needs at least one reference.

  1. Lactadherin as a biomarker of tumor subtype, progression and metastasis

Comment 3: “…placenta, endometrium, ovaries and fallopian tubes [22-26].” – so I didn’t find specific ref for placenta, ovaries and fallopian tube (except proteinatlas.org).

22.Bocca, S.M. et al., Milk fat globule epidermal growth factor 8 (MFG-E8): A novel protein in the mammalian endometrium with putative roles in implantation and placentation. Placenta 2012, 33(10), 795–802. – endometrium

23.Arklie, J.; Taytor-Papadimitrou, J.; Bodmer, W.; Egar, M.; and Millis, R. Differentiation antigens expressed by epithelial cells in the lactating breast are also detectable in breast cancer. Int. J. Cancer 1981, 28, 23-29. – Only normal uterus mentioned, other is about breast and breast cancer. Some other normal tissue e.g. liver, kidney, pancreas, mentioned.

24.Ceriani, R.L.; Peterson, J.A.; Blank, E.W. Breast Cancer Diagnosis with Human Mammary Epithelial Antigens and the Prospective Use of Antibodies against Them in Therapy. In: Honn KV, Powers WE, Sloane BF. editors. Mechanisms of Cancer Metastasis. Developments in Oncology. 1986; pp. 235-257. – could not find it in even google. Looked in ncbi.nlm.nih.gov and google.

26.Human Protein Atlas https://www.proteinatlas.org/ENSG00000140545-MFGE8/tissue. Accessed 13 May 2020. – shows expression of RNA and protein in all female reproductive organs.

4.1. State of art on lactadherin translational medicine research and intellectual property

Comment 4: “Finally, a recent patent postulates the detection of lactadherin, among other 2 proteins, as tumor markers for several types of cancer. Interestingly, it considers a method to detect these proteins in blood samples of cancer patients (CN106596940A). It proposes that the combined detection of these 3 proteins can enhance test sensitivity, specificity and precision. “

Please, mention those 2 other proteins - CA153 and CEA.

Comment 5: “Strikingly, despite the different patents and therapeutic approaches, to date, lactadherin has not been a focus of investigation in clinical trials. Interestingly, based on studies by Touhy and collaborators (Mazumder’s group) [57,58], a phase I clinical trial begin the enrollment of subjects to test safety, as well as the most effective dose of an alpha-lactalbumin vaccine to treat patients with non-metastatic TNBC in October, 2021. This is interesting because, as lactadherin, alfa-lactalbumin is not expressed in normal tissues at immunogenic levels (only in lactation), but is overexpressed in emerging tumors. Currently, this clinical trial is recruiting patients and the estimated primary completion date is May, 2022 (https://clinicaltrials.gov/ct2/show/NCT04674306).

Can authors elaborate a bit more on why we still do not have clinical trials with lactadherin antibodies?

Comment 6: “On the other hand, another recent patent (CA2988771A1) includes the use of lactadherin to purify a subpopulation of EVs, based on the interaction between this protein and PS present on EVs surface.”

Please, add that PS is - phosphatidylserine (PS) binding agent. Also, grammar inaccuracy (type-o) in word “interaction”.

  1. Extracellular vesicles and exosomes as promotors of breast cancer metastasis

Comment 7: “Breast cancer- associated fibroblasts secrete exosomes that have been shown to promote tumor mobility, invasion and dissemination of breast cancer cells through the Wnt-planar cell polarity pathway (Wnt-PCP pathway) [68].”

  1. Joyce, D.P.; Kerin, M.J.; and Dwyer, R.M. Exosome-encapsulated microRNAs as circulating biomarkers for breast cancer. Int J Cancer 2016, 139(7), 1443-8.

This is a review article in which I didn’t find the corresponding statement. Perhaps it is some type-o, because the proof for the statement is in reference 69.

  1. Role and use of lactadherin in EVs and

Comment 8: Moreover, lactadherin presence in EVs have been recently reported in several studies [73,74], which has led to consider it as a possible new marker of EVs.

  1. Novikova, S. et al., 2020 Proteomic Approach for Searching for Universal, Tissue-Specific, and Line-Specific Markers of Extracellular Vesicles in Lung and Colorectal Adenocarcinoma Cell Lines. Int. J. Mol. Sci 2020, 21(18), 6601.

This is review that mentions Lactadherin in one sentence.Perhaps 84. Veron et al, would be better reference.  

  1. Pathan, M.; Fonseka, P.; Chitti, S.V.; Kang, T.; Sanwlani, R.; Van Deun, J.; Hendrix, A.; and Mathivanan, S. Vesiclepedia 2019: a compendium of RNA, proteins, lipids and metabolites in extracellular vesicles. Nucleic Acids Res. 8;47(D1), D516-D519. Accessed September 17th, 2021

I could not find neither in article nor in http://microvesicles.org/query_results good proof of the given statement.

Comment 9: “On the other hand, using bioluminescent and fluorescent labeling of EVs, an interesting recent work by Gupta and collaborators have shown that the proportion of lactadherin+ EVs is significantly lower than that of tetraspanin+ EVs. Sorting of the EVs based on GFP showed a high percentage of tetraspanin+ EVs (15-25%). In contrast, just 1% of the sorted EVs was lactadherin+, which could have implications on EVs isolation or affinity purification protocols [81].”

  1. Gupta, D. et al. Quantification of extracellular vesicles in vitro and in vivo using sensitive bioluminescence imaging. J Extracell Vesicles 2020, 9(1), 1800222.

I agree that it is an interesting study, however, authors must state that: (i) study was performed on cell lines, not only human, but mouse as well; and (ii) – “1% of the sorted EVs was lactadherin+” - it was observed only in HEK cell line, (normal cell line, not tumour!). Please, revise the sentence and be precise.

Comment 10: “With all these background, now the questions are, on one hand is lactadherin present in exosomes or EVs secreted by tumor cells? – can authors re-phase this question? Do they mean either exosome or EVs, or they are using it as two different terms? The same comment applies for latter marts in that paragraph, where authors talk about EVs and/or exosome.

Comment 11: “As lactadherin have several functions both, as a cell adhesion molecule, and triggering intracellular signaling cascades, it is possible to think that it could have similarl roles as part of EVs’ membranes of EV’s cargo.”  – wording problem - perhaps better is …as part of EVs membrane or EVs cargo.

Comment 12: In figure 2A. please correct the name “Macropho” to macrophage. In addition, it would be good to put word “Lactadherin” on the bottom and put an arrow to that molecule.

Author Response

Thank you very much for your kind and assertive advice and comments to improve our review “Lactadherin: from a well-known breast tumor marker to a possible player in extracellular vesicle-mediated cancer progression”. This is the detail of the corrections we have made in each of the 17 comments.

  1. Scrupulous checking of the references used in this manuscript, there are few comments/suggestions mentioned below.

Answer: We have checked and corrected all references thoroughly.

  1. One main comment about style – to look again at the manuscript and minimize use of word “interesting study” or “interestingly”. There are just a few places where it is not necessary.

Answer: We have decreased the use of the words and phrases “interestingly” and “interesting study”. We have replaced some of them for other words and connectors duly marked in the text.

  1. In addition, please, revise the references on figures in the text. Some unneeded (e.g. Fig 1. mention is not needed in Introduction paragraph 2 in sentence “However, although lactadherin multifunctionality is still under discussion and analysis (Fig. 1), most authors agree that lactadherin is projected as a strong focus of investigation and a promising therapeutic target and/or biomarker for cancer and other diseases [10].”) or not in the order (e.g. Fig.3. is mentioned in the text ahead of Fig.2. – revise, if it is really needed).

Answer: We eliminated the mentioned reference to figure 1. Also, we have corrected the order of figures 2 and 3 references in the text.

  1. Comment 1: “Other important functions described for lactadherin include antiviral properties, modulation of the innate immune system through the activation of tumor associated macrophages [8], promotion of VEGF-induced angiogenesis, and tissue regeneration [9,10,11-15].” Could not find in given references statement about antiviral properties.

Answer: We appreciated this comment since the role of lactadherin as a protein with antiviral properties should be strengthened, therefore we include 2 references about lactadherin antiviral roles (refs 8,9)

  1. About VEGF-induced angiogenesis authors gives reference 12 and 13, however, both are just mentioning original paper by [20] - Silvestre JS, Thery C, Hamard G, et al. Lactadherin promotes VEGF-dependent neovascularization. Nat Med 2005;11:499, hence, please, review necessity for reference 12. and 13. Matsuda, A. et al., Milk fat globule EGF factor VIII ameliorates liver injury after hepatic ischemia-reperfusion. Journal of Surgical Research 2013, 180, Pages e37-e46. Deng, Ke-Diong. et al., Restoration of Circulating MFGE8 (Milk Fat Globule-EGF Factor 8) Attenuates Cardiac Hypertrophy Through Inhibition of Akt Pathway. Hypertension 2017, 70, 770–779.

Answer: It is appreciated how compromising the revision of the references is, and how it has shown involuntary errors like these. In this way, we have checked both references and the one by Silvestre et al. We have corrected the order of these references. Silvestre et al., now is ref 11. Matsuda et al., and Deng et al., are now refs 57,58. These two studies are included as other diseases where lactadherin have been targeted.

  1. In addition, reference 14 is about mesenchymal stem cells. Yes, in this review, there is a reference to Motegi et al., original paper about VEGF relation to MEG8. Please, cite the original paper. Motegi, S.; Ishikawa, O. Mesenchymal stem cells: The roles and functions in cutaneous wound healing and tumor growth. Journal of Dermatological Science 2017, 86, 83-89.

Answer: Thanks again for this comment because it has prompted us to look for the authors who have initiated the research, not only those who write the reviews. Therefore, the original paper by Motegi was also included. Both papers are now in refs 15,16.

  1. Comment 2: Statement “Moreover, recent studies have shown lactadherin presence in extracellular vesicles (EVs) derived from cancer cell lines and in cancer patient specimens, which was associated with tumor aggressiveness and worst prognosis” – needs at least one reference.

Answer: References 23 and 24 were rearranged in the text (Lobos et al., and Crescitelli et al.,).

  1. Lactadherin as a biomarker of tumor subtype, progression and metastasis

Comment 3: “…placenta, endometrium, ovaries and fallopian tubes [22-26].” – so I didn’t find specific ref for placenta, ovaries and fallopian tube (except proteinatlas.org).

22.Bocca, S.M. et al., Milk fat globule epidermal growth factor 8 (MFG-E8): A novel protein in the mammalian endometrium with putative roles in implantation and placentation. Placenta 2012, 33(10), 795–802. – endometrium. 23.Arklie, J.; Taytor-Papadimitrou, J.; Bodmer, W.; Egar, M.; and Millis, R. Differentiation antigens expressed by epithelial cells in the lactating breast are also detectable in breast cancer. Int. J. Cancer 1981, 28, 23-29. – Only normal uterus mentioned, other is about breast and breast cancer. Some other normal tissue e.g. liver, kidney, pancreas, mentioned. 24.Ceriani, R.L.; Peterson, J.A.; Blank, E.W. Breast Cancer Diagnosis with Human Mammary Epithelial Antigens and the Prospective Use of Antibodies against Them in Therapy. In: Honn KV, Powers WE, Sloane BF. editors. Mechanisms of Cancer Metastasis. Developments in Oncology. 1986; pp. 235-257. – could not find it in even google. Looked in ncbi.nlm.nih.gov and google. 26.Human Protein Atlas https://www.proteinatlas.org/ENSG00000140545-MFGE8/tissue. Accessed 13 May 2020. – shows expression of RNA and protein in all female reproductive organs.

Answer: It is appreciated how thorough the review of the references is and the context in which they are used. This way, first of all, the study by Bocca et al., include the detection and characterization of endometrial and placental tissue.

The studies by Arklie et al., and Ceriani et al., describes the presence of lactadherin in breast tissue. Finally, both ovaries and fallopian tubes were checked again in protein atlas database.

Ovaries were deleted from the text because lactadherin expression is low in healthy ovarian tissue (according to this database). Fallopian tubes were correctly identified as a tissue which has high levels of lactadherin expression (at mRNA and protein levels). We have attached the precise link in the ref of Human Protein Atlas database (now ref 29) where we can check lactadherin levels in female tissues.

  1. 1. State of art on lactadherin translational medicine research and intellectual property

Comment 4: “Finally, a recent patent postulates the detection of lactadherin, among other 2 proteins, as tumor markers for several types of cancer. Interestingly, it considers a method to detect these proteins in blood samples of cancer patients (CN106596940A). It proposes that the combined detection of these 3 proteins can enhance test sensitivity, specificity and precision. “

Please, mention those 2 other proteins - CA153 and CEA.

Answer: To be more precise and leave no room for doubt both proteins, CA153 and CEA were included in the text.

  1. Comment 5: “Strikingly, despite the different patents and therapeutic approaches, to date, lactadherin has not been a focus of investigation in clinical trials. Interestingly, based on studies by Touhy and collaborators (Mazumder’s group) [57,58], a phase I clinical trial begin the enrollment of subjects to test safety, as well as the most effective dose of an alpha-lactalbumin vaccine to treat patients with non-metastatic TNBC in October, 2021. This is interesting because, as lactadherin, alfa-lactalbumin is not expressed in normal tissues at immunogenic levels (only in lactation), but is overexpressed in emerging tumors. Currently, this clinical trial is recruiting patients and the estimated primary completion date is May, 2022 (https://clinicaltrials.gov/ct2/show/NCT04674306). Can authors elaborate a bit more on why we still do not have clinical trials with lactadherin antibodies?

Answer: As we can see in some of the studies mentioned in the review, anti-lactadherin antibodies have been used to target breast and other cancer approximately since the 90s. Also, in the section about intellectual property, we saw that the use of lactadherin antibodies and other derivatives have been legally protected. Some patents from 2008-2009 “blocked” the use of anti-lactadherin antibodies, blocking peptides and other molecules. We can even see that in those years the research about lactadherin as an antitumor therapy markedly decreased.  We have talked with lawyers and professionals in the area of intellectual property and clinical trials in our university and they say that it is possible that the broad claims of these patents could have slowed lactadherin research and even possible funding for clinical trials intended to use anti-lactadherin antibodies. Maybe, when these patents expire, it might be possible to develop clinical trials in this area with anti-lactadherin antibodies. As this is just our hypothesis, we don’t include this in the main text.

  1. Comment 6: “On the other hand, another recent patent (CA2988771A1) includes the use of lactadherin to purify a subpopulation of EVs, based on the interaction between this protein and PS present on EVs surface.” Please, add that PS is - phosphatidylserine (PS) binding agent. Also, grammar inaccuracy (type-o) in word “interaction”.

Answer: Despite the fact that we have already define the abbreviation for phosphatidylserine (PS) in page 2, we include the correction. We also checked the type-o error in “interaction”.

  1. Extracellular vesicles and exosomes as promotors of breast cancer metastasis

Comment 7: “Breast cancer- associated fibroblasts secrete exosomes that have been shown to promote tumor mobility, invasion and dissemination of breast cancer cells through the Wnt-planar cell polarity pathway (Wnt-PCP pathway) [68].” Joyce, D.P.; Kerin, M.J.; and Dwyer, R.M. Exosome-encapsulated microRNAs as circulating biomarkers for breast cancer. Int J Cancer 2016, 139(7), 1443-8. This is a review article in which I didn’t find the corresponding statement. Perhaps it is some type-o, because the proof for the statement is in reference 69.

Answer: The reviewer is right. We have corrected this reference mistake.

  1. Role and use of lactadherin in EVs and

Comment 8: Moreover, lactadherin presence in EVs have been recently reported in several studies [73,74], which has led to consider it as a possible new marker of EVs. Novikova, S. et al., 2020 Proteomic Approach for Searching for Universal, Tissue-Specific, and Line-Specific Markers of Extracellular Vesicles in Lung and Colorectal Adenocarcinoma Cell Lines. Int. J. Mol. Sci 2020, 21(18), 6601. This is review that mentions Lactadherin in one sentence.Perhaps 84. Veron et al, would be better reference. Pathan, M.; Fonseka, P.; Chitti, S.V.; Kang, T.; Sanwlani, R.; Van Deun, J.; Hendrix, A.; and Mathivanan, S. Vesiclepedia 2019: a compendium of RNA, proteins, lipids and metabolites in extracellular vesicles. Nucleic Acids Res. 8;47(D1), D516-D519. Accessed September 17th, 2021

I could not find neither in article nor in http://microvesicles.org/query_results good proof of the given statement.

Answer: The study by Novikova et al., is not a review. It is an original paper in which the authors showed that lactadherin (also known as MFGE8) is present in EVs secreted by the H23 lung cancer cell line. They also mention that MFGE8 (lactadherin) have been shown to be present in different EVs according with the information in databases such as ExoCarta, a database very similar to Vesiclepedia. To be more clear, we included the exact link to Vesiclepedia webpage in which we can see that there are a lot of studies in which it is shown that lactadherin is present in human EVs (http://microvesicles.org/gene_summary?gene_id=4240). Also included in the references section (now ref 80). We also included the study by Veron et al., in this section.

  1. Comment 9: “On the other hand, using bioluminescent and fluorescent labeling of EVs, an interesting recent work by Gupta and collaborators have shown that the proportion of lactadherin+ EVs is significantly lower than that of tetraspanin+ EVs. Sorting of the EVs based on GFP showed a high percentage of tetraspanin+ EVs (15-25%). In contrast, just 1% of the sorted EVs was lactadherin+, which could have implications on EVs isolation or affinity purification protocols [81].” Gupta, D. et al. Quantification of extracellular vesicles in vitro and in vivo using sensitive bioluminescence imaging. J Extracell Vesicles 2020, 9(1), 1800222. I agree that it is an interesting study, however, authors must state that: (i) study was performed on cell lines, not only human, but mouse as well; and (ii) – “1% of the sorted EVs was lactadherin+” - it was observed only in HEK cell line, (normal cell line, not tumour!). Please, revise the sentence and be precise.

Answer: The reviewer is correct. We checked the information again and include some sentences to be as accurate as possible. We have corrected this directly in the text (section 6, page 10).

  1. Comment 10: “With all these background, now the questions are, on one hand is lactadherin present in exosomes or EVs secreted by tumor cells? – can authors re-phase this question? Do they mean either exosome or EVs, or they are using it as two different terms? The same comment applies for latter marts in that paragraph, where authors talk about EVs and/or exosome.

Answer: Probably this is a misunderstanding. As we mentioned, exosomes as a specific type or subpopulation of EVs. EVs are mainly composed by exosomes, microvesicles and apoptotic bodies. They differ each other in their origin or biogenesis. In particular, exosomes have an endosomal origin. So, we marked the difference when we write about exosomes in particular, or EVs as in general because, strictly speaking, exosomes are part of EVs (not the same). We have checked the references and include some additional information regarding molecular markers and another reference (ref 66) to be more accurate in the differences and definitions.

  1. Comment 11: “As lactadherin have several functions both, as a cell adhesion molecule, and triggering intracellular signaling cascades, it is possible to think that it could have similar roles as part of EVs’ membranes of EV’s cargo.” – wording problem - perhaps better is …as part of EVs membrane or EVs cargo.

Answer: Thank you very much for the revision in the use of the language. The wording problem has been checked.

  1. Comment 12: In figure 2A. please correct the name “Macropho” to macrophage. In addition, it would be good to put word “Lactadherin” on the bottom and put an arrow to that molecule.

Answer: Checked and corrected. Also, the figure was improved according with the reviewer comments.

Reviewer 2 Report

The manuscript “Lactadherin: from a well-known breast tumor marker to a possible player in extracellular vesicle-mediated cancer progression” is covering in details data about lactadherin, a promising marker not only for breast cancer but also for other types of cancer. Authors also collected the recent information about this protein in connection to its presence in extracellular vesicles derived from cancer cell lines and cancer patients. The review is professionally and well written. Illustrations are very informative and attractive.  The manuscript can be published in the journal “IJMS”.

Author Response

Thank you very much for your positive evaluation to improve our review “Lactadherin: from a well-known breast tumor marker to a possible player in extracellular vesicle-mediated cancer progression”